# Peer Academic Supports for Success (PASS) for College Students with Mental Illness: Open Trial

**DOI:** 10.3390/healthcare10091711

**Published:** 2022-09-07

**Authors:** Maryann Davis, Dori S. Hutchinson, Paul Cherchia, Laura Golden, Emily Morrison, Amanda Baczko

**Affiliations:** 1Transitions to Adulthood Center for Research, Department of Psychiatry, UMass Chan Medical School, Worcester, MA 01655, USA; 2Center for Psychiatric Rehabilitation, Sargent College of Health and Rehabilitation Sciences, Boston University, Boston, MA 02215, USA; 3Population and Quantitative Health Science Department, UMass Chan Medical School, Worcester, MA 01655, USA

**Keywords:** college students, youth mental health, peer coaching, academic supports, accommodations

## Abstract

Increasing numbers of college students have serious mental health conditions, but their dropout rates are high and debt accrual is common. A well-specified intervention that colleges can directly offer their undergraduates with serious mental health conditions that sustains their academic persistence is greatly needed. The Peer Academic Supports for Success (PASS) coaching model was developed to address this need. This study’s goal was to conduct an open trial of the initial PASS model to test the feasibility of the model and research methods in preparation for more rigorous testing. Ten college juniors and seniors, with and without lived mental health experience, were hired, trained, and supervised to be PASS peer coaches. Twelve undergraduate students with academically impairing mental health conditions served as study participants and received PASS. Student data were collected at baseline and two semesters post baseline. Intervention feasibility data were assessed through coach report. Results indicate PASS can be delivered with fidelity by peer coaches, can attract and retain students, and is safe. Results also suggest that PASS has significant effects on most of the targeted proximal outcomes. The PASS findings are promising as a college-based intervention to support young adult students with mental health conditions.

## 1. Introduction

Individuals with 4-year college degrees earned 1.6 times the amount that individuals with a high school diploma did in 2020 in the United States [1]. A bachelor’s degree also affords disproportionate protection during economic downturns [2]. Parents from different socioeconomic and racial groups alike hope that their children will attain college degrees (e.g., [3,4,5,6]). Parent financial contributions towards their children’s college education are currently the largest source of students’ college funding in the U.S., surpassing scholarships, grants, and student loans [7]. Thus, their children’s success in achieving a college degree is both an aspiration and financial investment for many families.

Rates of mental health conditions among 4-year college students are high [6,8,9,10,11,12], their dropout rates are high [13,14,15], among the highest of any disability groups [16], and debt accrual is common [17]. Secondary students with mental health disabilities have the lowest grade performance of any disability group [16]. Thus, college success is more likely with academic supports. Colleges typically provide their students with academic supports including disability services. However, students with serious mental health conditions are unlikely to consider themselves in need of accommodations [16] or to utilize them [18]. Traditionally aged (i.e., ages 18–24) college students with mental health conditions use accommodations and disability services even less than older students [19]. Providing accessible, appealing academic supports should enhance attainment of a bachelor’s degree. 

Supported Education (SEd) interventions are designed to assist people with psychiatric disabilities take advantage of opportunities within postsecondary educational environments [20]. Outcome studies of SEd have examined interventions that served subgroups of undergraduates: mature adults in which the average age of study participants was over 30 years (e.g., [21,22,23,24,25]), veterans [26], or those with early psychosis (e.g., [27,28,29]). There are also published SEd outcomes for traditionally aged students (e.g., [30,31,32]). However, in all these studies, with rare exception, SEd was provided by a mental health or vocational rehabilitation service, or on a college campus, for students at a variety of nearby campuses. There is moderate evidence for the efficacy of all these models to improve academic engagement [28,33]. 

What is absent from this literature is an evidence-based SEd model designed for 4-year colleges to provide directly to their traditionally aged undergraduates. Such a model would address most public and private non-profit degree-granting postsecondary institutions in the U.S. (71.2% are 4-year institutions; [34] Table 317.10), and most of their students (75.2% of their undergraduates are ages 18–24; [35] Table 303.50). Such an SEd model would be a stellar contribution to the field. 

Theoretical Framework: Social Cognitive Career Development Theory (SCCDT) [36] is a comprehensive theory that has been consistently validated in college studies (e.g., [37,38,39]). It is based on the central concept of self-efficacy from Social Cognitive Theory [40]. Self-efficacy is an individual’s belief in one’s own capacity to execute behaviors necessary to produce specific performance attainments. In SCCDT, academic self-efficacy enhances motivation to pursue career or academic goals. Learnings from academic experiences and attainment are the strongest influence on future self-efficacy. SCCDT suggests that SEd interventions that enhance skills and increase positive academic experiences will enhance academic self-efficacy, which in turn, will enhance motivation for and persistence in academic goals. In college students with and without disabilities, progressing and remaining in college is bolstered by executive function skills [41,42,43], resilience [44,45,46], and self-determination skills [47,48] and hindered by weak social support [49]. Two executive functions, time management skills and goal setting, have shown immediate impact on college grades [50,51,52]. Recently, Jeffries and Salzer [53] found college students with mental illness to have lower academic self-efficacy and poorer study habits, many of which reflected weaker executive function skills, than students in the general population.

The Peer Academic Supports for Success (PASS) coaching SEd program for traditionally aged freshmen and sophomores with serious mental health conditions targets skills and capacities associated with progressing and remaining in college, to enhance positive experiences and self-efficacy and improve academic performance and persistence (see Figure 1). Coaching is a “goal-directed, results-oriented, systematic process in which one person facilitates sustained change in another” [54]. Coaching is a psychiatric rehabilitation-provider skill for assisting people with mental illness to learn and use skills and supports in relationship to a goal [55,56]. 

PASS was adapted from two sources; a student coach approach that excluded coaches with lived experience and was designed for freshman with autism spectrum disorders [57], and a professional coaching service provided by the Center for Psychiatric Rehabilitation at Boston University for students with serious mental health conditions. PASS was a stakeholder-engaged project that was guided by a national advisory board of 8–12 young adults with lived experience, and co-investigators who were recent young adult college students with lived experience, in partnership with three college faculty members. This formed the PASS Project Team. Project team young adults strongly advocated for PASS to be delivered by peers. Generally, peer-delivered interventions have been found to be efficacious for individuals with mental illness (e.g., [58,59,60,61,62]). However, the project team concluded that examining the capacity of “peers” (i.e., fellow undergraduate students) both with and without direct lived experience was important. Thus, PASS coaches include students with and without lived experience.

An initial step to refining PASS was to conduct a small open trial to examine the feasibility of PASS and its research methods that would allow the project team to identify problems and fix the approach and research methods in preparation for a more rigorous trial. 

## 2. Materials and Methods

The national youth advisory board that advised the PASS team advised multiple research projects in the research center. Members were recruited through social media and self-identified as having a serious mental health condition. Members were selected to represent diversity in race, ethnicity, LGBTQ community, gender, educational attainment, geographic regions, and systems involvement. The board was co-led by a young adult research staff member with lived experience and an external board member. The board reviewed this project multiple times in developing the grant proposal, annually during the study, and ad hoc to help with specific issues, such as recruitment methods, intervention questions, or interpretation of findings.

Site. The study site was a 4-year private not-for-profit “Doctoral University with Very High Research Activity” (Carnegie classification) in the Northeast region of the U.S., with a 25% admission rate requiring strong academic skills, and an undergraduate enrollment of approximately 18,000 students. The racial distribution of U.S. students was: 47% White, 19% Asian, 14% Latino, 5% Black, and 14% unknown, and 21% of undergraduates were foreign students. The majority (98%) were full-time students and lived in dorms (73%).

The institutional review board of the first author’s institution approved this research. The IRB for the study site ceded oversight to the first author’s IRB.

### 2.1. Participants

Students who received PASS (i.e., students) and PASS peer coaches (i.e., coaches) served as study participants. Student eligibility criteria included a self-reported mental health condition from the following DSM-V diagnostic categories: schizophrenia or other psychotic disorders, bipolar and related disorders, depressive disorders, anxiety disorders, trauma- and stressor-related disorders, feeding and eating disorders, attention deficit and disruptive behavior disorders, or obsessive-compulsive disorder. Additional eligibility criteria were: ages 18–24 years; self-report mental health treatment needs; self-report of mental health impediment to academic progress; full-time undergraduate freshman or sophomore enrolled at the University; English fluency; and able to give consent. Eligibility criteria for coaches were: ages 18–24 years; an undergraduate beyond 2nd-year status enrolled at the University; English fluency; able to give consent.

Students were recruited via flyers posted on campus and university websites and provided to university referral sources. Flyers contained links to a website where interested individuals completed an online preliminary screen that queried age, academic status at the University, English fluency, and self-report mental health diagnosis. Those that preliminarily screened in were invited to complete written informed consent for the final screen. Informed consent covered the procedures involved, the voluntary nature of participation, the minimal risks and rewards of participation, confidentiality and their limits, and assured them that their choice to participate or not would not affect any other services they received. The final screen consisted of the Brief Symptom Inventory [63,64] and a 21-item questionnaire developed for this study to assess academic difficulties due to their mental health. This Educational Barriers Questionnaire contained 17 items such as “Managing academic stress” and “Forming relationships with other students in my classes” for which students indicated the amount of difficulty they’d had, due to their mental health, in the most recent academic year, endorsed on a 4-point scale from “a lot of difficulty” to “no difficulty”. One additional item, “I tend to be absent from class due to my mental health condition(s).” was scored on a 4-point scale from “not at all” to “a lot”. Three yes/no items, queried withdrawing from courses, being put on academic probation, or taking a leave of absence, due to their mental health. Eligibility was met with a clinical cutoff score for the Brief Screening Inventory and at least one of the following; obtaining a score of 34 or higher on the first 18 items of the Educational Barriers Questionnaire or having been put on academic probation or taking a leave of absence due to their mental health. Students that completed the final screen were compensated with a $10 gift certificate.

Twelve students expressed interest in study participation. All completed the preliminary and final screen and were deemed eligible and invited to enroll in the study. Informed consent covering the same topics as for the final screen, but including the procedures for being in the open trial, was conducted and obtained from all subjects involved in the study. Written informed consent was conducted prior to baseline interview. All students consented to participate in the study and were assigned a PASS coach. 

The coach jobs were posted on the University student job board, and invitations to nominate coaches were sent to campus organizations and department faculty. The supervisor was a professional coach, licensed mental health counselor, and academic instructor employed by the University. The supervisor conducted interviews with applicants and selected the ten most qualified (e.g., previous peer coaching experience, displayed emotional maturity). While not a requirement that coaches had lived experience with a mental health condition, several self-identified as such. Written informed consent was conducted by a research team member.

#### Research Attrition 

Of the 12 students, all completed baseline surveys, and 11 completed the final survey. Analyses of proximal outcomes are based on these 11 individuals. Students completed 96% of all scheduled surveys. Data on intervention attrition, fidelity, grades, and school persistence are presented for the full sample. 

### 2.2. PASS Procedures

PASS Coaching was provided to all students. Students can receive coaching for up to two semesters. Students completed a form about their interests and why they wanted peer coaching, which the Supervisor used to match students and coaches based on shared interests and gender as much as possible (male coaches were only assigned to male students). Coaches initiated contact to begin coaching. Coaching was offered at least once/week. Students and coaches could meet in-person or via video. Texting was encouraged to facilitate brief check-ins or supportive messages. Coaches used action-oriented, goal-driven, and solution-focused coaching methods to conduct hands-on, collaborative and interactive work with students. The overall goal was for students to identify their academic challenges, and goals that would address them, then engage in activities to achieve the goals.

To help students identify goals, coaches could discuss the reasons students wanted coaching on the “interest” form or ask students about struggles they were having in classes, in completing assignments, or in academic performance. Coaches could suggest goals listed in the coach manual or share the list with the student for their choosing. Each goal had a described set of activities to address it. Each session included identifying goals and tasks to be worked on that session. Coaches were trained to encourage students to identify their desired goal, and to prioritize goals and choose the activities together. Goals and activities were reviewed in weekly supervision. 

Before working with students, coaches underwent extensive training to ensure they felt supported and confident in their coaching role. They received 12 h of formal training during the first 4 weeks of the fall semester. Training covered the specifics of the coaching role, campus resources, skills such as listening, responding, empathy, crisis response, suicide prevention and motivational interviewing. Training also covered many potential coaching activities. Training used a mix of webinar and in-person formats and included extensive role-playing. The supervisor provided one-hour weekly group supervision with peer coaches throughout the academic year. In addition to addressing immediate needs to ensure coaches were delivering PASS appropriately, supervision extended the initial training. Coaching skills were introduced in supervision based on the time of the semester, or common or specific needs of students. For example, time management activities and planning worksheets were practiced in supervision and then suggested to students to help prepare for final exams. The senior PASS co-developer supervised the supervisor weekly.

### 2.3. Data Collection Procedures

Student participants completed an assessment at baseline and end of Semester 2 (T2). Assessments were administered via a RedCap™ web-based database. Baseline assessments measured demographic and background factors. Baseline and T2 assessments measured proximal outcomes. A researcher sent students links to the survey and scheduled a time to complete it when the researcher was available to answer questions as needed. Students were given $40 gift cards for completing baseline assessments and $20 gift cards for T2 assessments. 

PASS coaches completed weekly activity logs. 

#### Measures

**Demographic and Background Measures**. Gender, age, LatinX ethnicity, race, class (i.e., freshmen, sophomore), number of courses currently enrolled in, residence (on-campus, off-campus not family, with family), history and current mental health treatment, and self-report of mental health diagnoses by a provider were measured at baseline via a standard questionnaire used in previous research (Author et al., 2015). 

**Implementation Measure-Fidelity.** PASS fidelity was conceptualized using Kutash and colleagues’ [65] approach to measuring fidelity. Adherence was measured through coach weekly activity logs that included each coach/student appointment, the goals focused on the activities engaged in and duration and type of contact. Coaching activities were limited to those listed in the manual. A session was defined as at least 50 min in face-to-face or video interaction that included at least one goal and one activity. Fidelity variables consisted of the proportion of weeks in which coaches meet minimum dose requirements (proportion of 2-week intervals with at least one session) and the proportion of appointments in which at least one goal was addressed and one coach activity conducted.

**Distal Outcomes Measures.** The two distal outcomes were assessed with formal records: each semester’s grade point average (GPA) for the two semesters following baseline, and persistence in their academic goal of completing college (i.e., being enrolled throughout the 3rd semester following baseline). No prior college grades were available for freshmen; thus, semester GPA was assessed at the end of Semester 1 and Semester 2. 

**Proximal Outcomes Measures.** Self-report measures were used to assess the proximal outcomes of PASS (see Figure 1). 

*Executive Function Skills.* These were measured with the total score on the Time Management Behavior Scale [66], a 33-item scale with items scored on a 5-point scale of “seldom true” to “very often true” [42]. Wording on the items was adapted to reflect school tasks rather than work tasks. This scale has good internal reliability (α = 0.71–0.91) [67,68,69,70]. 

*Resilience.* Resilience was measured with the total score on the Brief Resilience Scale (BRS), a 6-item scale scored on a 5-point scale, with good internal reliability (0.70< α <0.95) [71,72]. The BRS assesses the ability to bounce back from stress [73]. Reduction in experienced distress was assessed with the total score on the Kessler Psychological Distress Scale (K10) [74], a 10-item questionnaire scored on a 5-point Likert-type response scale. It measures distress based on questions about anxiety and depressive symptoms experienced in the past 4 weeks. It has good internal reliability (0.78 < α < 0.90) [75,76,77].

*Academic Self-Efficacy*. Academic self-efficacy was assessed using the total score on the Self-Efficacy for Broad Academic Milestones Scale (SE-Broad; [78]). It is comprised of 12 items; 5 on self-efficacy for completing academic milestones and 7 on coping with barriers related to academic success. Internal consistency estimates of 0.85 and above are reported [79,80]. General self-efficacy was assessed using the total score on the General Self-Efficacy Scale, a 10-item scale scored on a 4-point scale with good internal consistency (0.75 < α < 0.91) [81,82,83].

*Self-Determination*. Broad self-determination skills were assessed with the total score on the AIR Self-Determination Scale, a measure based on self-determined learning theory developed for secondary students [84,85]. It consists of 4 sections with 6 items in each, scored on a 5-point scale. We modified items to reflect the status of college students/young adults (e.g., “at school” was changed to “in my academic life”). It has acceptable internal reliability (0.74 < α < 0.918) [84,86]. *Self-empowerment skills* (a subset of self-determined behavior) was assessed with the total score on The Self-Determination and Self-Advocacy Questionnaire, a 16-item questionnaire scored on a 6-point scale. It assesses high school students’ skills in understanding and communicating their needs to individuals and knowing how to get what they need related to employment and postsecondary education goals [87]. Items were modified to focus specifically on goals for success at college. For example, for “I can list and discuss the academic accommodations I need to be successful in high school”, “high school” was changed to “college”. Five items were dropped that related to only to work or high school (e.g., their Individualized Education Plan) yielding an 11-item assessment. Self-empowerment skills were also assessed with the total score on The Patient Self-Advocacy Scale, a 12-item scale scored on a 5-point scale. It was developed for individuals with serious medical conditions and designed to assess the dimensions of (a) illness and treatment education, (b) assertiveness in health care interactions, and (c) potential for nonadherence [88]. Items were modified for (a) mental health conditions and (b) to reflect educational rather than medical supports. For example, “I am more educated about my health than most US citizens.” was changed to “I am more educated about my mental health than most college students.” It has acceptable internal reliability (0.78 < α < 0.83) [88,89]. *Help-Seeking* (a subset of self-determined behavior) for both academic help and mental health needs were assessed with a modified version of The General Help-Seeking Questionnaire [90]. It consists of 20-items scored on a 7-point scale that query the likelihood that an individual will seek help from 10 specific individuals (e.g., parent, doctor,) or from no one, for personal/emotional problems (10 items). The suicidal thoughts section (10 items) was dropped. This measure has good internal consistency (0.84 < α < 0.85) [90,91]. It was modified so that the 10 individuals reflected individuals found on campuses (e.g., roommate, counselor on campus) and retained those in their personal life (e.g., family members, religious leader). The total score on this section was used. A 10-item section was added that queried help-seeking for “an academic-related problem”. The total score on each section was used. 

*Social Support.* School-based social support was measured using the Sarason Social Support Questionnaire (SSSQ6; [92]). For each of the 6 items, respondents indicate the number of people available to provide support in each of 6 areas, then rate their overall level of satisfaction with the support given in each area. The total support score and the total satisfaction score were used. Responses were limited to listing individuals who could provide social support to the participant while living at the university. The SSSQ6 has high internal reliability (0.90 < α < 0.93) [92,93]. 

### 2.4. Analysis Plan

Given the small sample size, medians and ranges were examined to explore participant characteristics, and characteristics of the PASS intervention: participant attrition, duration and amount of intervention received, and intervention fidelity. Non-parametric tests were used to examine outcome variables over time. Wilcoxon signed-rank tests were used to compare baseline and T2 scores on measures of proximal outcomes, and to compare Semester 1 to Semester 2 grades. 

## 3. Results

### 3.1. Participant Characteristics

As can be seen from Table 1, the majority of participants were white, non-LatinX women. Their mental health conditions were reflected in the large proportion taking medications and the majority receiving some type of mental health counseling or therapy. 

### 3.2. Intervention Feasibility

#### 3.2.1. Attrition 

Coaching was offered for two semesters, until the final week of classes in the spring semester. Dropping out was defined as the last coaching session occurring before the final 3 weeks of the second semester. No student explicitly requested to stop receiving coaching prior to the end of the spring semester. However, three students disengaged (i.e., declined to respond to repeated attempts to schedule a coaching session) prior to the final 3 weeks of the spring semester. This disengagement was deemed attrition from the program, yielding an attrition rate of 25%. 

#### 3.2.2. Intervention Amount and Safety

The median number of coaching sessions attended was six (range 3–13). The median % of weeks that students met with coaches was 60.1% (range 15.8–90.9%) during “active engagement”. Active engagement was the time period between first coaching appointment and the end of the second semester, except for the three students who ended early. For those students, the weeks from the first to last session were used as their period of active engagement. In weeks in which students met with PASS coaches, the median average time spent with a PASS coach was 1.11 hrs. (range 0.71–1.63). PASS was offered with complete safety. Participants reported no adverse events during follow-up surveys, and the coach supervisor reported no known adverse events. 

#### 3.2.3. Intervention Fidelity

Fidelity criteria for frequency of meeting with each student was set a priori at a minimum of one session (of at least 50 min) every other week while school was in session. From their extensive professional coaching experience with this population, the lead PASS developers suggested this minimum, though weekly sessions were encouraged. The PASS project team agreed with this criterion. A session was defined as addressing at least one goal (from the list of possible goals) and conducting at least one coaching activity (from the list of allowable activities). Any appointments in which this criterion was not met were not counted as a session. The median proportion of weeks per student in which this criterion was met during active engagement was 0.9 (range = 0.38–1.00). However, only half of student/coach pairs met this criterion for 100% of weeks, four student/coach pairs met the criteria for 75–80% of weeks, and two student/coach pairs met criteria less than 51% of weeks. 

Goals and activities that were addressed were examined in appointments of ≥50 min. PASS coaches reported an appropriate student goal for 91% of appointments, and all of these appointments also included at least one activity. The median number of student goals per appointment was two (range = 0–5). Appointments that did not include at least one goal and one activity clustered in one coach, for whom only 56% of appointments included a goal and an activity. The median proportion of appointments that met criteria for the remaining coaches was 1.0 (range 0.83–1.0). 

Effective time management was the most common goal students selected to work on, and academic self-efficacy was the least common (See Table 2). The median number of activities conducted per appointment was four (range = 0–11). Only one session for one coach included no activities. The most common activity conducted during sessions was working with students to identify their values, interests, and strengths, and the least common was showing them emotional agility apps (See Table 2). 

### 3.3. Intervention Outcomes

#### 3.3.1. Distal Outcomes

The distal outcome of persistence was defined as completing the third semester following baseline. All participants completed the third semester from baseline. Median GPA for Semester 1 was 3.24 (range 2.53–3.93) and for Semester 2 was slightly higher at 3.28 (range = 1.60–4.0). Though 8 of 12 students had an increase in their GPA from Semester 1 to Semester 2, a related-samples Wilcoxon signed-rank test revealed no significant difference (Statistic = 40.50, *p* = 0.91).

#### 3.3.2. Proximal Outcomes

A related-samples Wilcoxon signed-rank test revealed significant change from baseline to the end of the second semester in several variables (See Table 3). Time management, resilience, self-efficacy, and self-determination measures showed significant improvements from baseline to T2. Change in academic self-efficacy scores approached significance (*p* = 0.074), as did the Patient Self-Advocacy Scale (*p* = 0.068). Likelihood to seek help for academic or mental health problems both declined significantly from baseline. There were no significant changes in any other measures.

## 4. Discussion

This SEd intervention has the potential to be the first evidence-based intervention that 4-year colleges can directly implement for their own traditionally aged undergraduates. Given that the majority of institutions of higher education in the U.S. are 4-year colleges, and the majority of their undergraduates are traditionally aged, if subsequently proven effective, PASS has the potential to help the majority of undergraduates with mental health disabilities pursuing bachelor’s degrees to succeed in that quest. The central findings of this initial feasibility study of PASS are positive and justify subsequent rigorous testing. The findings demonstrate that the intervention is feasible. Student coaches with and without lived experience were trained to provide PASS coaching to fidelity. Students enrolled in PASS and the retention rate was acceptable. No safety issues arose. Research feasibility findings were also encouraging. The fidelity measure captured adequate variability. However, fidelity assessment would be strengthened by a student-completed fidelity tool. Research retention was strong. Students completed outcome measures and transcripts were successfully obtained. The selected measures showed good variability (i.e., there were no ceiling or floor effects and a range of scores were observed) with results generally detecting change. While the sample size was miniscule, several outcomes were notable. Change in the distal outcome of school persistence was uniformly positive. Moreover, pre-post changes in proximal outcomes were positive and statistically significant for executive function skills, resiliency, self-efficacy, and self-determination. 

The fidelity findings indicated that nine of the ten coaches consistently met the concrete standards for fidelity in which the overwhelming majority of their appointments with students met criteria for fidelity: meeting for at least one session every two weeks that was at least 50 min. in duration and included addressing at least one of the list of possible student goals and engaging in at least one of the list of specific activities. This resulted in the majority of students consistently receiving coaching with fidelity. Clearly, the training and supervision structure is adequate to support good coaching. Yet, given the one coach who did not perform the majority of appointments with fidelity, and the majority of sessions for two students were not conducted with fidelity, future supervision protocols need to be able to correct weak coaching more quickly. Overall, however, these findings demonstrate that student coaches can be trained to deliver the PASS intervention with fidelity. 

Importantly, PASS was also delivered safely, as demonstrated by the complete absence of any adverse events recorded either through data collection or by supervisor report. The 75% intervention completion rate was lower than desirable, though perhaps not uncommon for college-based programs. It is interesting that the majority of students chose to focus on time management skills and resiliency skills. This likely reflects both the major challenge of college (i.e., completed assignments and preparation for exams on time) and a major source of stress (i.e., performance). However, the most common activity, identifying student values, interests, and strengths, addressed academic self-efficacy skills rather than time management skills. Coaches may have been using these activities to also build rapport. Conversely, students may have identified the goal of time management, but coaches may have subsequently felt the need to bolster the student by engaging in these activities. Generally, future PASS supervision protocols should emphasize the better matching of activities and goals. 

Informational focus groups with the students and coaches, conducted by the research team at the end of the academic year, explored what PASS aspects worked and didn’t. Students described that conversations with their coaches felt directionless at times. Coaches described that they hesitated to lead because that would not be “student led”. Similarly, coaches noted great difficulty in scheduling sessions when students felt overwhelmed. As a result, the PASS developers specified how to initiate conversations that explicitly invite students to consider potential topics, and to set goals more formally with their coaches so that both are clear about the focus of their work together. A student workbook was also developed so that students could more easily consider potential goals and activities. They have also structured coaching sessions to include scheduling the next session. Initial sessions and subsequent sessions as needed will include discussing anticipated reticence to meet when overwhelmed, identifying when that is likely, and discussing how coaching remains helpful during those times. These changes to the manuals should be included in future testing of this intervention. 

Biebel and colleagues [94] described the value of peers with lived experience delivering supports to college students with mental health conditions in their college focus groups that queried important elements of SEd. Corrigan and colleagues [95], in their qualitative study about potential strengths and challenges of peer academic coaching, found many anticipated benefits, but also many anticipated challenges: fear of burning out student-coaches with mental illness, concerns about identifying good peer coaches and providing sufficient supports, and concern about the school’s liability. The PASS model, guided by partners who are young adults with lived experience, defined “peers” first as fellow students, and having lived mental health experience as desirable but not mandatory. Future investigation of this model will benefit from identifying differential engagement, retention, and outcomes in students by coaches with and without lived experience.

While statistically significant pre-post changes with such a small sample in an open trial are not expected, they were found in six measures of proximal outcomes. The significant changes in executive function skills, resilience, self-efficacy and self-determination were in the desired direction. The unexpected reduction in academic and mental health help-seeking may reflect that PASS reduced these needs. In addition, the changes in median score and range were in the right direction for four of the remaining six measures. Distal outcome findings were also encouraging with academic persistence (being enrolled the third semester after baseline) uniformly positive, and the end of Semester 1 to end of Semester 2 changes in grades, though not significant, in the right direction. Overall, these findings support advancing to more rigorous investigation of the efficacy of PASS.

Given that the essential purpose of feasibility trials is to refine the intervention and research methods, study limitations include the absence of randomization or a control group. The absence of a control group limited examination of grades since freshmen had no baseline. Significant impact on grades at the end of each semester would be detectable with a control group as comparison. The absence of a control group also limited understanding the relative strengths or weaknesses of the outcomes; however, that would be the main goal of a larger randomized trial. The absence of randomization procedures limited the ability to predict the impact of randomization on recruitment and retention. An additional limitation is that because participants provided a self-reported professional diagnosis, the presence of a mental health condition may not be accurate; students may have mis-remembered a professional’s report of their diagnosis, and the clinical diagnosis they report may not have been rigorously assessed. However, the high prevalence of both current treatment and mental health medication lends confidence to the presence of some mental health condition. The sample also does not reflect students who are unaware of having a mental health condition. The sample does, however, reflect students who would likely self-refer for coaching. Lastly, this research was conducted at a highly competitive private not-for-profit university. Future research should include other types of colleges.

## 5. Conclusions

Four-year colleges do not currently have access to evidence-based SEd that they can directly provide to support the academic success of their traditionally aged undergraduates with mental health conditions. If eventually proven effective, PASS will be the first such intervention. This open trial was the initial step towards developing such an intervention. The trial produced positive indications for further testing of PASS. Some adjustments are needed to the intervention, to better guide students and coaches in co-developing goals and conversations and increase session frequency.

## Figures and Tables

**Figure 1 healthcare-10-01711-f001:**
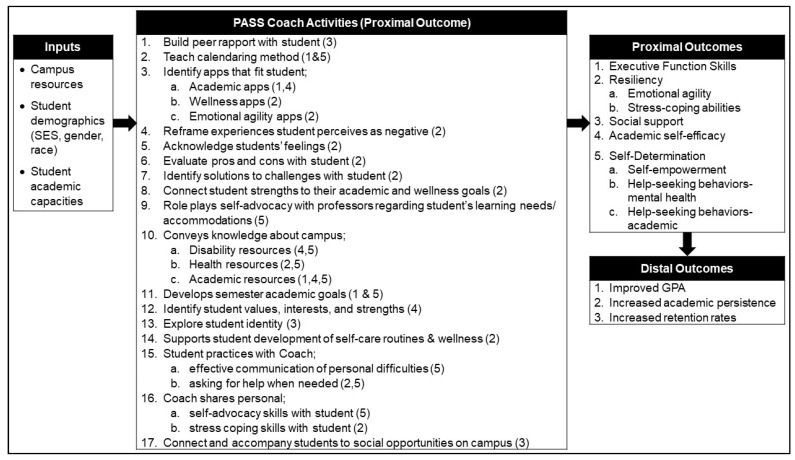
Logic model of the Peer Academic Supports for Success (PASS) coaching model.

**Table 1 healthcare-10-01711-t001:** Baseline characteristics of study participants (N = 12).

*Variable*	*n=*	*Variable*	*n=*
**Gender**		**Race**	
Man	3	White	9
Woman	8	Black	0
Other	1	Asian	4
**Age**		Multiple	1
18	3	**# Courses enrolled in**	
19	3	One	1
20	3	Two	1
21	1	Four	5
22	1	Five	2
23	1	Six	3
**LatinX**	0		
**Residence**		**Current Self-Report Diagnoses**	
On-campus housing	8	Major Depressive Disorder	3
Off-campus housing	3	Bipolar Disorder	2
Parent’s home	1	Generalized Anxiety Disorder	5
**Mental Health Treatment History**		Obsessive-Compulsive Disorder	1
Yes	10	Schizoaffective Disorder	1
No	2	Eating Disorder	1
**Current Counseling/Therapy**		Borderline Personality Disorder	1
Yes	9	Other MH Disorder	3
No	3	Multiple MH Disorders	3
**Current Medications**		Attention Deficit Disorders	3
Yes	5	Autism Spectrum Disorder	1
No	7	Learning Disability	2

**Table 2 healthcare-10-01711-t002:** Proportion of sessions in which specific goals for sessions were identified and specific PASS coach activities occurred for 10 coaches reporting on 89 sessions.

	Proportion of Sessions
**Student Goal**	
Effective time management	0.57
Resiliency	0.44
Self-advocacy	0.36
Knowledge of Tools/Resources	0.30
Academic self-efficacy	0.21
**Coach-Student Activity**	
Identify student values, interests, and strengths	0.55
Reframes experiences	0.52
Calendaring	0.44
Self-care routines/habits	0.38
Academic goals	0.37
Academic resources	0.31
Personal self-advocacy skills	0.30
Communicate personal difficulties	0.29
Stress coping skills	0.25
Asking for help	0.17
Other	0.17
Disability resources	0.10
Role plays self-advocacy conversations	0.10
Academic apps	0.06
Wellness apps	0.06
Emotional agility apps	0.01

**Table 3 healthcare-10-01711-t003:** Baseline to end of Semester 2 (T2) comparisons of instrumental outcomes for n = 10 students receiving PASS coaching.

*Variable and Measure*	*Baseline*		*T2*		*Statistic*	*p*=
	Median	(Range)	Median	(Range)		
**Executive Function Skills**						
Time Management Behavior Scale, total score	85	(64–100)	90	(68–111)	48.0	0.037
**Resilience**						
Brief Resilience Scale, total score	16	(11–24)	18	(13–23)	35.0	0.136
Kessler Psychological Distress Scale (K10), total score	27	(12–42)	24	(10–36)	6.0	0.028
**Academic Self-Efficacy**						
Self-Efficacy for Broad Academic Milestones Scale, total score	75	(30–95)	75	(50–96)	45.0	0.074
General Self-Efficacy, total score	26	(15–34)	30	(22–40)	66.0	0.003
**Self-Determination**						
AIR–Self Determination Scale, total score	63	(51–109)	74	(60–120)	53.5	0.008
**Self-Determination–Self-Empowerment**						
Self-Determination and Self-Advocacy Skills Questionnaire, total score	35	(15–66)	34	(25–66)	38.0	0.656
Patient Self-Advocacy Scale, total score	39	(31–50)	43	(35–49)	53.5	0.068
**Self-Determination–Help-Seeking**						
Help Seeking Questionnaire–Academic section score	35	(21–64)	31	(22–46)	9.0	0.033
Help Seeking Questionnaire–Mental Health section score	35	(16–70)	30	(15–50)	0.0	0.005
**Social Support**						
Sarason Social Support Questionnaire–total support score	2.33	(0–9)	2.83	(0–6.67)	26.0	0.678
Sarason Social Support Questionnaire–total satisfaction score	3.17	(1–6)	2.67	(1–6)	10.5	0.292

## Data Availability

The data presented in this study are available on request from the corresponding author. The data are not publicly available due to the small sample size precluding assurance of confidentiality.

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
