# Peer review of "Peer Academic Supports for Success (PASS) for College Students with Mental Illness: Open Trial"

_healthcare, 2022, doi:10.3390/healthcare10091711_

Round 1

Reviewer 1 Report

Dear authors, 

greetings of the dya, with interest i read your manuscript and found it very interesting,

however there are some points to review:

11-     Ameliorate the quality of the figure 1 because it is  blurry

22-     You chooses your participants by self-reported impairing mental health conditions: It may create a bias because many subjects may be sick but not knowing they have a mental disease

33-     Ethical considerations must be reported in the Subject and methods section

44-     References: Many so old references ( 2-15, 17-22, 24-30, 33-46, 48, 49, 51-5, 57-74)

please update

Best regards and best of luck

Author Response

Thank you for your review of our manuscript. We have responded, below, to each suggestion.

1. Ameliorate the quality of the figure 1 because it is  blurry

Response: we have imported a powerpoint slide object that has larger font in hopes that the image is not blurry.  If this paper is accepted, we will work with journal staff to ensure a clear image.

2. You chooses your participants by self-reported impairing mental health conditions: It may create a bias because many subjects may be sick but not knowing they have a mental disease

Response: Helping undiagnosed students to get an accurate mental health diagnosis is beyond the scope of the study or the intervention being examined. We chose self-identification because it reflects the practices of most university services and would thus, realistically, be the students that use PASS when it is not in research. However, we have added an acknowledged limitation of self-reported diagnosis in the limitations section of the discussion.   

3. Ethical considerations must be reported in the Subject and methods section

Response: We have added ethics details in the second and third paragraphs of page 4.

4. References: Many so old references ( 2-15, 17-22, 24-30, 33-46, 48, 49, 51-5, 57-74) please update.

Most of these older references are correct.  They either are the appropriate citation for a well-established measure (e.g. references 57-74), are describing the history of supported education work (which started in the 1980’s), the specific work this intervention is based on, or simply are the most recent findings.   We have reviewed and added more recent literature where we could find it. All remaining older references remain as the cited literature because nothing more recent has been published that we could find, or there is limited research and inclusion is still important.

Thank you.

Reviewer 2 Report

Dear authors, thank you for submitting the manuscript: Peer Academic Supports for Success (PASS) for College Students with Mental Illness: Open Trial. This study is interesting since it approaches such an important aspect as mental health in university students.

Some specific comments:

-In the introduction you should delete the lines 30-32.

-Lines 51-60 "e.g." is indicated on several occasions but no further information is provided. Please provide clarification.

-The results sections should be improved, as they are not very well explained.

-I recommend improving the discussion and include more of your research results.

Thank you!

Author Response

Thank you for your review of our paper.  The following are our responses to your helpful suggestions.

1. In the introduction you should delete the lines 30-32.

Response: Done

2. Lines 51-60 "e.g." is indicated on several occasions but no further information is provided. Please provide clarification.

Response: The examples now follow each “e.g.,”

3. The results sections should be improved, as they are not very well explained.

Response: we have added details and reviewed and edited for clarity.  We hope this has improved the presentation of our results.

4. I recommend improving the discussion and include more of your research results.

Response: we have added discussion of more of the results.

Thank you for these suggestions.

Round 2

Reviewer 2 Report

Dear Authors.

Thank you for making the suggested changes.

Author Response

Thank you for your review.